# Increased Response to Immune Checkpoint Inhibitors with Dietary Methionine Restriction in a Colorectal Cancer Model

**DOI:** 10.3390/cancers15184467

**Published:** 2023-09-07

**Authors:** Lauren C. Morehead, Sarita Garg, Katherine F. Wallis, Camila C. Simoes, Eric R. Siegel, Alan J. Tackett, Isabelle R. Miousse

**Affiliations:** 1Department of Biochemistry and Molecular Biology, University of Arkansas for Medical Sciences, Little Rock, AR 72205, USAgargsarita@uams.edu (S.G.); ajtackett@uams.edu (A.J.T.); 2Department of Pathology, University of Arkansas for Medical Sciences, Little Rock, AR 72205, USA; ccsimoes@uams.edu; 3Department of Biostatistics, University of Arkansas for Medical Sciences, Little Rock, AR 72205, USA

**Keywords:** methionine restriction, immune checkpoint inhibitors, PD-L1, STING, colorectal

## Abstract

**Simple Summary:**

In colorectal cancer cells, reducing the essential amino acid methionine in the culture medium led to an increase in two markers of response to immune checkpoint inhibitors: MHC-I and PD-L1. The increase in MHC-I was associated with an increase in STING and type I interferon. The increase in PD-L1 was associated with an increase in type II interferon. Reducing methionine in the diet led to an increase in PD-L1 membrane expression in the tumors and a better response to immune checkpoint inhibitors in a mouse model of colorectal cancer.

**Abstract:**

Dietary methionine restriction (MR), defined as a reduction of methionine intake by around 80%, has been shown to reproducibly decrease tumor growth and synergize with cancer therapies. In this study, we combined DMR with immune checkpoint inhibitors (ICIs) in a model of colon adenocarcinoma. In vitro, we observed that MR increased the expression of MHC-I and PD-L1 in both mouse and human colorectal cancer cells. We also saw an increase in the gene expression of STING, a known inducer of type I interferon signaling. Inhibition of the cGAS–STING pathway, pharmacologically or with siRNA, blunted the increase in MHC-I and PD-L1 surface and gene expression following MR. This indicated that the cGAS–STING pathway, and interferon in general, played a role in the immune response to MR. We then combined dietary MR with ICIs targeting CTLA-4 and PD-1 in an MC38 colorectal cancer tumor model developed in immunocompetent C57BL/6 mice. The combination treatment was five times more effective at reducing the tumor size than ICIs alone in male mice. We noted sex differences in the response to dietary MR, with males showing a greater response than females. Finally, we observed an increase in membrane staining for the PD-L1 protein in MC38 tumors from animals who were fed an MR diet. MHC-I was highly expressed in all tumors and showed no expression difference when comparing tumors from control and MR-treated mice. These results indicated that MR increased PD-L1 expression both in vitro and in vivo and improved the response to ICIs in mice.

## 1. Introduction

Methionine is an essential amino acid used for protein synthesis and as a precursor for the antioxidant glutathione and the major cellular methyl donor S-adenosylmethionine. In healthy animals, reducing the dietary intake of methionine by about 80%, termed dietary methionine restriction (MR), is associated with an increase in lifespan and an improvement of glucose and lipid regulation [1,2,3,4,5,6,7]. In tumor-bearing animals, dietary MR decreases tumor growth and improves the response to radiation therapy [8,9], chemotherapy [10,11,12,13,14], and targeted therapy [15]. Recent evidence suggested that dietary MR can also improve the response to immune checkpoint inhibitors (ICIs), but the mechanism underlying this effect has not been fully investigated [16,17].

Since their introduction in 2011, ICIs have rapidly changed the cancer field. ICIs are antibodies that target immune molecules involved in dampening the anti-tumor immune response, the most common being CTLA-4, PD-1, and PD-L1. For specific types of tumors with a high tumor mutational burden, ICIs have increased the response rates, with remarkably few side effects. In metastatic colorectal cancer, anti-PD-1 therapy is approved as a first-line treatment for patients whose disease is characterized as either microsatellite instability-high or mismatch repair-deficient (MSI-H/dMMR). This subtype is found in 4–5% of patients with metastatic colorectal cancer [18]. An anti-PD-1/anti-CTLA-4 combination currently in clinical trials has already generated encouraging results in patients with MSI-H-dMMR colorectal cancer, although randomized studies are still required [19]. In MSI-H/dMMR rectal cancer, response rates up to 100% have been recorded [20]. However, the MSI-H/dMMR subtype represents only a small fraction of all colorectal cancers. For the majority of the patients, the efficacy of ICIs is limited by the poor immunogenicity of their tumors [21]. 

One important way by which tumors escape the immune system is by downregulating the expression of the major histocompatibility complex I (MHC-I) at their surface [22,23]. MHC-I presents intracellular antigens to the immune system, including abnormal proteins produced by the tumor cells. In humans, the complex consists of four domains; three domains from the α chain (encoded by HLA-A, HLA-B, and HLA-C in human) and one encoded by β_2_-microglobulin. During its translocation to the membrane, other events such as folding, stabilization, and autophagy can influence the final abundance of MHC-I presented at the surface [24].

Another marker that predicts the response to immune checkpoint inhibitor is PD-L1. PD-L1 expression by cancer cells downregulates the anti-tumor immune response by binding to the inhibitory PD-1 receptor on cytotoxic T-cells. Most immunotherapy regimens include an anti-PD-1/PD-L1 antibody, alone or in combination with CTLA-4. High expression of PD-L1 in tumors indicate a reliance on this inhibitory mechanism and therefore represents one of the factors that affect the tumor susceptibility to anti-PD-1 therapy [25]. Several factors regulate PD-L1 expression, including interferon signaling [26].

Previous research showed that restricting methionine in the diet modulates the immune system [4,27]. In healthy animals, dietary MR delays the age-related decline in immunity. At 18 months of age, mice fed a methionine-restricted diet showed a pattern of T-cell subsets in the peripheral blood that was closer to the one of young animals [4]. However, T-cell differentiation is also reported to depend on methionine uptake [28]. Because of this conflicting activity of methionine in the immune system, we investigated whether MR would have a positive or a negative impact on the response to ICIs and the mechanisms underlying this effect. 

Our results indicated that MR increased MHC-I through type I interferon and cGAS–STING signaling and PD-L1 through type II interferon signaling in colorectal cancer cells (Figure 1). Using a mouse model of colorectal cancer, we showed that MR improved the response to immune checkpoint inhibitors in mature males but not in mature females. The membrane expression of PD-L1 was increased in tumors from males consuming an MR diet.

## 2. Materials and Methods

### 2.1. Cell Culture

HT29 (CVCL_0320) cells were obtained from ATCC (Manassas VA, USA) and used between passages 4 and 12. B16F10 OVA and MC38 (CVCL_B288) cells were a gift from Dr. Alan Tackett. The cells were maintained in standard DMEM with L-glutamine, 4.5 g/L glucose, and sodium pyruvate (Corning, Corning NY, USA), supplemented with 10% FBS (Corning) and 100 IU penicillin and streptomycin (ThermoFisher Scientific, Waltham MA, USA). The methionine-free and control media were prepared from high-glucose, no-glutamine, no-methionine, no-cystine DMEM (ThermoFisher Scientific) supplemented with 10% dialyzed serum (12–14 kD) (R & D Systems, Minneapolis, MN, USA), 100 IU penicillin and streptomycin (ThermoFisher Scientific), 4 mM L-glutamine, and 1 mM sodium pyruvate (ThermoFisher Scientific). L-cystine (Millipore-Sigma, Burlington, MA, USA) was resuspended in 1M HCl and added to the cell media at a final concentration of 150 µM. For the control medium, L-methionine (Millipore-Sigma) was resuspended in PBS and added to the cell medium at a final concentration of 200 µM. For the methionine-restricted medium, L-methionine was added at a final concentration of 5 µM. Unless otherwise indicated, the assays were performed after 24 hours of treatment. For cGAS-STING inhibition, the STING inhibitor C-176 was used at a final concentration of 5 µM (Tocris Bioscience (Bio-Techne, Bristol, UK), and the cGAS inhibitor RU.521 was used at a concentration of 1 µM (MedChemExpress, Monmouth Junction, NJ, USA); the treatments were performed for 24 hours in both cases. The JAK inhibitor ruxolitinib was used at a final concentration of 4 µM (ThermoFisher Scientific).

### 2.2. Gene Expression Analysis

To measure gene expression, we plated the cells in 6-well plates in triplicates in standard DMEM medium and let them attach. We then rinsed the cells in PBS and replaced the cell culture medium with the treatment media, as described above. RNA was extracted using QIAzol lysis reagent (QIAGEN, Germantown, MD, USA), following the manufacturer’s instructions. RNA quality and quantity were assessed by spectrophotometry (Nanodrop, ThermoFisher Scientific). Reverse transcription was performed on 1 µg of purified RNA using the iScript Reverse Transcription Supermix (Bio-Rad, Hercules, CA, USA). For each quantitative real-time PCR reaction, 20 ng of cDNA was amplified with the iTaq Universal SYBR Green Supermix (Bio-Rad), using technical duplicates. Fold changes in expression were calculated using the ΔΔC_t_ method with the mean of the technical duplicates for each biological triplicate [29]. We used the internal control gene *Rplp0*. The primers used are listed in Appendix A.

### 2.3. Protein Expression Analysis

To measure protein expression, we plated the cells in 10 cm dishes and proceeded as described in “Gene expression analysis”. After incubation in the treatment media, the cells were rinsed with PBS and detached with trypsin. The cells were spun at 1000× *g* for 5 min at 4 °C. The cell pellet was then rinsed once with PBS and resuspended in RIPA buffer supplemented with a protease inhibitor cocktail (Millipore-Sigma) and phosphatase inhibitor tablets (ThermoFisher Scientific) for 30 min at 4 °C with agitation. The resulting lysate was spun at 10,000× *g* for 10 min at 4 °C. The protein concentration was measured in the resulting supernatant using a BCA method (ThermoFisher Scientific). Immunoblotting was performed by applying the quantified protein lysate to a Bis-Tris gel (NuPAGE, ThermoFisher Scientific) and then transferring the proteins unto a PVDF membrane. The membrane was blocked in EveryBlot blocking buffer (Bio-Rad) and probed with an anti-STING antibody (1:1000) overnight at 4 °C (#50494, Cell Signaling Technology, Danvers, MA, USA).

### 2.4. Flow Cytometry

To measure the expression of MHC-I and PD-L1 on the cell surface, we plated the cells in 6-well plates in triplicates, as described above. The cells were harvested with trypsin and rinsed, and 1 million cells were resuspended in PBS containing 1% fetal bovine serum (ThermoFisher Scientific). Fluorescently-labeled antibodies against MHC-I (Cat# 311406 and 114608, PE-labeled), H-2K^B^ bound to SIINFEKL (Cat# 141604, PE-labeled), and PD-L1 (Cat# 393610 and 124312, APC-labeled) (Biolegend, San Diego, CA, USA) were added at a final dilution of 1:100. The cells were counter-stained with DAPI at a final concentration of 1 ng/mL and analyzed on a BD LSR Fortessa instrument (BD Biosciences, Franklin Lakes, NJ, USA). We included compensation controls. The resulting data were processed using the software FlowJo (BD Biosciences). We filtered the cells based on side and forward scatter to select for intact, single cells. We then excluded dead cells through DAPI staining. Few cells were DAPI-positive at the time point studied. The remaining cells were analyzed for MHC-I and PD-L1 surface expression. For the assessment of reactive oxygen species, the cells were treated with CTL and MR media, as above. After 24 h, the plated cells were incubated with CellROX Deep Red reagent (Cat# C10422, ThermoFisher) at a final concentration of 2 µM in PBS for 30 min at 37 °C. The cells were then harvested and analyzed as described above.

### 2.5. Animal Studies

This project was approved by the Institutional Animal Care and Use Committee at the University of Arkansas for Medical Sciences (IACUC #4119). The C57BL/6 mice were purchased from Jackson Laboratories (Bar Harbor, ME, USA). The mice were kept under standardized conditions with controlled temperature and humidity and a 12 h light/dark cycle. The mice were 5 months of age at the beginning of the experiment and were fed an identical standard laboratory chow up to Day 5 of the experiment. 

We injected the mice subcutaneously with 4 × 10^5^ MC38 cells on the right hip (*n* = 8 animals per group). After 5 days of incubation, we changed the diet to a methionine-restricted (MR) diet containing 0.12% methionine or an otherwise identical control (CTL) diet containing 0.65% per weight methionine (Teklad Diets TD.190775 and TD.140520, Appendix A). We maintained these diets continuously until the endpoint. On days 5, 8, 11, and 13, the mice in 2 groups (CTL+ICI and MR + ICI) were injected intraperitoneally with monoclonal antibodies against PD-1 (RMP1-14, 250 μg per dose) and CLTA-4 (9D9, 100 μg per dose) (BioXCell, Lebanon, NH, USA) in sterile PBS or a control PBS solution [30]. We measured the tumors daily from Day 5, and calculated the tumor volumes using the formula V = (W^2^ × L)/2 as previously described [31,32]. When the tumors reached 500–600 mm^3^ or on day 35, we anesthetized the mice with isoflurane, and retroorbital bleeding was performed. We allowed the collected blood samples to coagulate at room temperature in serum gel tubes (Sarstedt, Newton, NC, USA) and then centrifuged them at 10,000× *g* for 10 min at 4 °C. The serum was transferred to new tubes and flash-frozen until analysis. The tumors were harvested and fixed in 10% formalin for immunohistochemistry or flash-frozen in liquid nitrogen. We repeated the experiment using only the diet under the same conditions to obtain material for IHC. 

### 2.6. Immune Cell Analysis

At sacrifice, we collected whole blood from the retro orbital sinus of tumor-bearing animals. We measured blood parameters with a Vetscan HM5 Hematology Analyzer.

### 2.7. Immunohistochemistry

To evaluate PD-L1, CD8, and MHC-1 expression in formalin-fixed paraffin-embedded tissue sections of mouse tumors, immunohistochemical staining was performed with conventional techniques using an avidin–biotin complex, diaminobenzidine chromogen, and hematoxylin counterstaining [33]. The sections (4 µm thick) were deparaffinized, rehydrated, and blocked for endogenous peroxidase, followed by incubation with rabbit anti-PD-L1 (64988, 1:100, Cell Signaling Technology), rat anti-CD8 (14-0808-80, 1:50, Invitrogen, Waltham, MA, USA), and rabbit anti-MHC-1 (NBP3-09017, 1:50, Novus Biologicals, Centennial, CO, USA) antibodies overnight at 4 °C. This was followed by a 30 min incubation with biotinylated goat anti-rabbit IgG (1:400, Vector Laboratories). The sections were subsequently incubated with the avidin–biotin–peroxidase complex for 45 min (1:100, Vector Laboratories). Peroxidase binding was visualized using a 0.5 mg/mL 3,3-diaminobenzidine tetrahydrochloride solution (Millipore-Sigma); the sections were then rinsed and counterstained with Harris’s hematoxylin, dehydrated, and coverslipped. The stained slides were scanned using an Aperio Scanner CS2 at 20×.

The slides were coded and randomly evaluated by one pathologist investigator with expertise in gastrointestinal pathology, who was not aware of the group assignment. The histological sections were examined using ×20, ×40, ×100, ×200, ×400 magnifications (Olympus BX50,Center Valley, PA, USA). The immune-mediated status based on PD-L1 staining in tumor cells and immune cells (lymphocytes and macrophages), CD8 positivity within tumor cells, and MHC-1 was assessed by blinded evaluation. 

PD-L1 expression was determined by using the combined positive score (CPS). The specimen were divided into 2 groups:

Score 0 = CPS < 1 (no PD-L1 expression)

Score 1 = CPS ≥ 1 (PD-L1 expression)

PD-L1 positivity was defined as PD-L1 expression on ≥1% of tumor (membranous staining) and/or immune cells (lymphocytes or macrophages with membranous and/or cytoplasmic staining of any intensity).
$$\mathrm{CPS}=\frac{\mathrm{Number}\;\mathrm{of}\;\mathrm{PD}-L1-\mathrm{stained}\;\mathrm{cells}\;\left(\mathrm{tumor}\;\mathrm{cells},\;\mathrm{lymphocytes},\;\mathrm{macrophages}\right)}{\mathrm{Total}\;\mathrm{number}\;\mathrm{of}\;\mathrm{viable}\;\mathrm{tumor}\;\mathrm{cells}}\times 100$$

### 2.8. Data Analysis

The software GraphPad Prism 8.3.0 (GraphPad Software, San Diego, CA, USA) was used to represent the data graphically and perform statistical analyses. Comparisons were made using an unpaired T test for pairwise comparisons and 2-way ANOVA with Šidák’s multiple comparison test (CTL vs MR within each treatment group) for experiments with a 2 × 2 design. SAS v9.4 (The SAS Institute, Cary, NC, USA) was used to analyze the tumor volumes vs. days post-injection, as follows: first, we log-transformed the tumor volumes to stabilize the variances and to facilitate statistical inference on ratio changes. If a mouse’s tumor volume was 0 mm^3^, then its log-volume was set equal to 0. Then, we used the MIXED Procedure in SAS v9.4 to analyze the longitudinal log-volume data from males and females, separately. For each sex, we modelled the data’s autocovariance structure with a heterogeneous 1st-order autoregressive model. We used the sex-specific models to estimate the mean log-volume with 90% confidence interval of each group on each post-injection day and to compare the MRD+ICI and CTL+ICI groups at unadjusted α = 0.05 for differences in their mean log-volumes on each day post-injection. Finally, we exponentiated the log-volume differences to obtain ratio changes in tumor volume and we also exponentiated the group mean log-volumes with 90% confidence intervals to display them graphically as tumor growth curves.

## 3. Results

### 3.1. Methionine Restriction Increases MHC-I Expression in Cancer Cells

MHC-I and PD-L1 are two markers of the response to ICIs. To assess the possible interaction between MR and immune checkpoint blockade, we measured the effect of MR on the expression of these two markers in vitro. We treated HT29 human colon cancer cells for 24 h in control DMEM medium containing 200 µM methionine or in an otherwise identical medium containing 5 µM methionine (methionine-restricted). We found that the expression of both *HLAA* and *CD274* genes (encoding the A subtype of MHC-I heavy chain and PD-L1, respectively) was significantly elevated in methionine-restricted cells compared to control cells (1.7×, *p* < 0.0001 and 1.6×, *p* = 0.0018 respectively) (Figure 2A,B). To elucidate whether this increase in gene expression was functionally relevant, we measured the corresponding protein surface expression by flow cytometry. We found that the surface detection of MHC-I and PD-L1 was indeed increased by MR (Figure 2C,D). To assess antigen presentation, we used murine B16F10 OVA cells, which express the SIINFEKL ovalbumin peptide. In that cell line, MR also increased the abundance of MHC-I at the cell surface (Figure 2E). Furthermore, SIINFEKL peptide presentation in the context of MHC-I at the surface of methionine-restricted cells was nearly double that of cells grown in control levels of methionine (Figure 2E). These results indicated that MR increased MHC-I gene expression and that this increase in gene expression was associated with an increase in the corresponding protein surface abundance in more than one cell line. Finally, MR also increased the display of antigens in the context of MHC-I on the cell surface in cancer cells.

### 3.2. The Increase in MHC-I Expression Is STING-Dependent

One of the triggers of MHC-I expression is the detection of cytosolic dsDNA through the cGAS–STING pathway. We hypothesized that inhibiting the cGAS–STING pathway would blunt the increase in MHC-I in response to low methionine levels. We observed that *STING* gene expression was elevated 1.6 times in methionine-restricted HT29 cells (*p* = 0.0012, Figure 3A). The STING protein level was also elevated in methionine-restricted cells. We then treated the cells with the STING inhibitor C-176 and analyzed *HLAA* gene expression. The increase in *HLAA* gene expression was blunted following STING inhibition, from 1.7 times (*p* = 0.0033) to 1.2 times (*p* = 0.0564) (Figure 3B). We found similar results when looking at MHC-I protein expression at the cell surface with flow cytometry (Figure 3C). The MHC-I increase in surface expression following MR was also blunted using the cGAS inhibitor RU-521 (Appendix A). The treatment with C-176 had less clear effects on the *CD274* gene (PD-L1) expression. C-176 itself led to an increase in *CD274* expression, without further increase when applying the combination of C-176 and methionine restriction. We therefore could not definitely conclude from this result that STING expression was associated with the increase in *CD274* (Figure 3D). We also observed an increase in gene expression for *HLAB*, *HLAC*, as well as interferon α and β (Appendix A). These results suggested that the increase in MHC-I expression in colon cancer cells following MR was mediated at least in part through the cGAS–STING pathway. 

### 3.3. Dietary Methionine Restriction Increases PD-L1 Expression in the MC38 Adenocarcinoma Model Cell Line

After establishing that MR elevated MHC-I expression in vitro, we investigated if this increase translated into an improvement in the response to ICIs in vivo. The MC38 murine colon adenocarcinoma cells are commonly used to model the effects of ICIs in humans because they are syngeneic in C57BL/6 and can therefore form tumors in immunocompetent animals. Similarly to the human colon cancer cell line, MR led to an increase in the MHC-I component gene *H2Kb* in murine MC38 cells in vitro (Figure 4A). However, when we analyzed the surface expression of MHC-I in MC38 cells, MR led to an unexpected decrease in MHC-I presence at the surface (Figure 4B). This was contrary to what we had observed in HT29 and B16F10 cells. For PD-L1, on the other hand, both gene expression and surface protein expression were significantly increased (Figure 4C,D). While STING signaling is dependent on type I interferon signaling, the regulation of PD-L1 is thought to be more closely related to type II interferon. We used the type I/II interferon signaling inhibitor Ruxolitinib. In this case, the increase in PD-L1 gene expression following MR was reduced by more than half. Ruxolitinib also blunted the increase in H2Kb gene expression (Appendix A). MR was also associated with an increase in various components of the interferon system (Appendix A). These results indicated that interferon signaling was involved in the increase in PD-L1 gene expression in response to MR. 

### 3.4. Dietary Methionine Restriction Improves the Response to ICIs In Vivo

We assessed the effect of dietary MR on the response to ICIs. We injected MC38 cells subcutaneously. The mice were 3 months of age at the beginning of the experiment. After 5 days, we replaced the animal diets with a control diet or an otherwise identical diet restricted in methionine. We initiated the therapy with ICIs concomitantly with the diet on day 5. The treatment with ICIs alone led to a 72–73% reduction in MC38 tumor volume in both females and males (Figure 5A). In males, the combination treatment led to a further reduction in tumor volume, the combined effect being 5× greater than that observed with ICIs alone (Figure 5A,B, Appendix A). However, we did not observe any improvement in the response to ICIs in females consuming a methionine-restricted diet (Figure 5A and Appendix A). We repeated the experiment in 3-month-old CTL and MR females for a total of 20 animals in each group and once again did not identify any difference in tumor volume between control and MR females (Appendix A). Previous results indicated a beneficial effect of MR in MC38 tumors in younger female mice [16], indicating that sexual maturity may influence the response to MR. To verify if age was a factor in the response to MR in females, we injected MC38 tumor cells in 6-week-old female mice. As before, the diets were initiated 5 days later. We found that the difference between control and MR female mice mirrored those found in males (Appendix A). These results support the idea that sexually mature females show little benefit from MR in terms of tumor volume, as opposed to young females and mature males. As expected from a relatively mild MR, we did not note any significant difference in body weight between groups (Figure 5C).

### 3.5. Dietary Methionine Restriction Increases PD-L1 Expression and CD8 Infiltration In Vivo

To confirm that the improvement in tumor control observed with the MR diet in males was also associated with an increase in MHC-I and PD-L1, we performed immunohistochemistry. We injected the animals with MC38 cells subcutaneously as before and initiated the diets after five days. We harvested and fixed the tumors when they reached a volume of 500–700 mm^3^. The tumor volumes in males and females were consistent with our previous observations. We did not measure any significant change in white blood cells or lymphocytes in association with the diets in animals of either sex (Appendix A). There was a significant decrease in hemoglobin in the methionine-restricted group in both females and males, with a small but significant decrease in red blood cells in females. To our knowledge, the hemoglobin values have not been previously reported in relation to MR, but our results are in line with data reported for complete methionine starvation in the presence of acute myeloid leukemia [34]. We then performed immunohistochemistry analysis on the tumor tissues. Our analysis did not identify any change in MHC-I staining or CD8 infiltration in the sections from animals consuming a methionine-restricted diet. However, we did observe an increase in PD-L1 staining associated with MR in males (Table 1). The mean combined positive score (CPS) for PD-L1 in the MR group was 15.2 and was significantly greater compared to the CPS of 6.2 in the control group (*p* = 0.016, unpaired *t*-test). This is in good agreement with the increase in PD-L1 gene and protein expression observed in vitro in MC38 cells after MR. We did not identify any change in PD-L1 staining in tumors from female animals. 

## 4. Discussion

The use of ICIs as a therapy for cancer has had a major impact on both patient survival and quality of life. Immune checkpoint blockade has been approved for a subtype of metastatic colorectal cancer with high tumor mutational burden. However, its efficacy is limited in more common subtypes with lower mutational burden. We described here that dietary MR improved the response to ICIs in a model of colorectal cancer. MR led to an increase in the expression of PD-L1, whose interaction with PD-1 is the main target of ICIs. This increase was seen both in vitro and in vivo. We also found an increase in MHC-I, which displayed antigens at the cell surface for surveying by the immune system, in the HT29 human colon cancer cell line in vitro. 

The increase in MHC-I expression translated into an increased display of antigens on the cell surface, as attested by the increase in the presence of the SIINFEKL peptide conjugated to MHC-I at the surface of murine B16F10 OVA cells. Our results indicated that the increase in MHC-I expression was mediated at least in part through the cGAS–STING pathway. STING inhibition with C-176 blunted the increase in MHC-I after MR, both at the gene expression level and at the surface protein expression level. A similar activation of the cGAS–STING pathway was described in response to a decrease in a different amino acid, i.e., arginine [35]. Furthermore, we also found evidence of type I interferon signaling (interferon α and β) in response to MR that supports our cGAS–STING observations. However, we cannot exclude the possibility that additional pathways may contribute to the observed effect. 

Although the expression of PD-L1 was slightly affected by blocking the cGAS–STING pathway, PD-L1 is more closely associated with type II interferon (ɣ) signaling [26,36]. In line with our results, recent evidence points to some degree of cross-talk between interferon ɣ and STING [37]. We used the JAK inhibitor ruxolitinib to demonstrate that blocking both type I and type II interferon signaling resulted in a blunting of the PD-L1 response to MR. We also observed an increase in the gene expression of the interferon gamma receptor and of STAT1 with MR, supporting the involvement of type II interferon and in line with previous results [26,38]. These results are also in good agreement with previous results showing an increased in interferon ɣ in methionine-restricted tumors [16]. We believe that additional factors such as NRF2 [39], already well described in the response to MR in cancer cells [40,41], may also contribute to the increase in PD-L1.

When we combined dietary MR with ICIs in the MC38 model, we observed a 5× improved response with the combination treatment compared to ICIs alone in males. This indicated that the changes that we observed in vitro might have a direct consequence in vivo on the response to therapy. However, there was no measurable benefits in females to either MR alone or the combination of MR and ICIs in relation to MC38 tumor growth. This was unexpected, as previously published data indicated a response to MR in young C57BL/6 female mice injected with MC38 cells [16], as well as in a melanoma model subjected to intermittent MR [42]. MC38 is a cell line derived from a female animal, which may be expected to raise a weaker immune response in females. However, the similarity in response to ICIs between males and females in the control diet argues against this. Li and colleagues [16] injected the cells in 7-week-old animals, while we performed the experiment in 3- to 5-months-old animals, as shown in Figure 4. Our data support the conclusion that young females do in fact respond to MR, contrarily to mature females. This suggests that sexual maturity influences the response to MR, in line with the reported effects of sex on antitumor immune responses (reviewed in [43,44]). We previously reported a similar sex difference in the way the gut microbiome responds to dietary MR in sexually mature animals [45]. More experiments are required to identify the direct contribution of different methionine levels on tumor growth and the host responsiveness and indirect effects, such as changes in glucose levels and microbiome composition. 

In our experiments, we measured an increase in MHC-I gene expression in both the human and mouse colorectal cancer cell lines after MR in vitro. However, this increase did not lead to a corresponding increase in surface protein expression in the murine MC38 model in vitro. We also did not measure any change in MHC-I protein expression in tumors from methionine-restricted animals. We believe that the discrepancy between gene expression and membrane abundance was most likely due to an increase in endocytosis and/or in ubiquitylation of the protein. On the other hand, we did observe that PD-L1 surface expression was increased in vitro by MR and that a larger proportion of methionine-restricted tumors showed high PD-L1 expression than control tumors. This is in opposition to a previous study reporting a paradoxical decrease in PD-L1 with MR in vitro, which the authors attributed to RNA methylation [16]. Unfortunately, that group did not report PD-L1 expression in the tumors. Importantly, there is a dual effect of PD-L1 on tumors. On the one hand, a high PD-L1 expression is a marker of responsiveness to immune checkpoint inhibitors. On the other hand, a high PD-L1 expression outside the context of immune checkpoint inhibitor therapy is usually a marker of poor immune response. PD-L1 dampens the anti-tumor immune response [46,47] and promotes DNA repair [48]. 

MR shares commonalities with caloric restriction and with compounds referred to as “caloric restriction mimetics”. One of these compounds is the anti-diabetic drug metformin. Metformin was previously shown to ameliorate the response to ICIs in animal models [49]. Interestingly, metformin was also reported to activate the cGAS–STING pathway in cancer cells [50]. Our own results showed that the caloric restriction mimetic and polyphenol resveratrol also increased the expression of genes via the cGAS–STING pathway. As previously mentioned, decreases in the amino acid arginine also led to the stimulation of the cGAS–STING pathway. The evidence suggests that this activation fits in a larger context of increased type I and type II interferon signaling. This also suggests that other related compounds and interventions related to caloric restriction may similarly impact interferon signaling and ultimately the response to ICIs. Finally, our data indicate that sexual maturity influences the response to MR in females, confirming previous reports in non-tumor models [45,51,52].

## 5. Conclusions

Our results revealed a previously undiscovered link between MR and the interferon pathway. We found novel effects of MR, which was shown to increase MHC-I and PD-L1 expression. In vivo, this translated into an improvement in the response to immune checkpoint inhibition in mature males and was associated with an increase in the membrane expression of PD-L1 in the tumors. Interestingly, tumors from immature females but not mature females responded to MR. These sex differences may be particularly relevant to the increasing population of premenopausal women with early-onset colorectal cancer. However, our results are limited to a murine preclinical model and remain to be confirmed in patients.

## Figures and Tables

**Figure 1 cancers-15-04467-f001:**
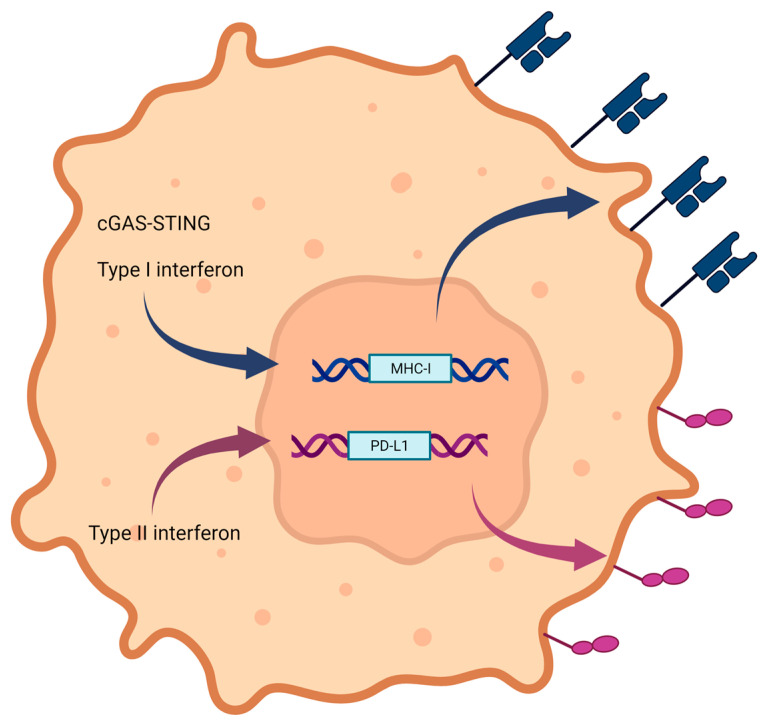
MR increases MHC-I and PD-L1 expression through interferon signaling.

**Figure 2 cancers-15-04467-f002:**
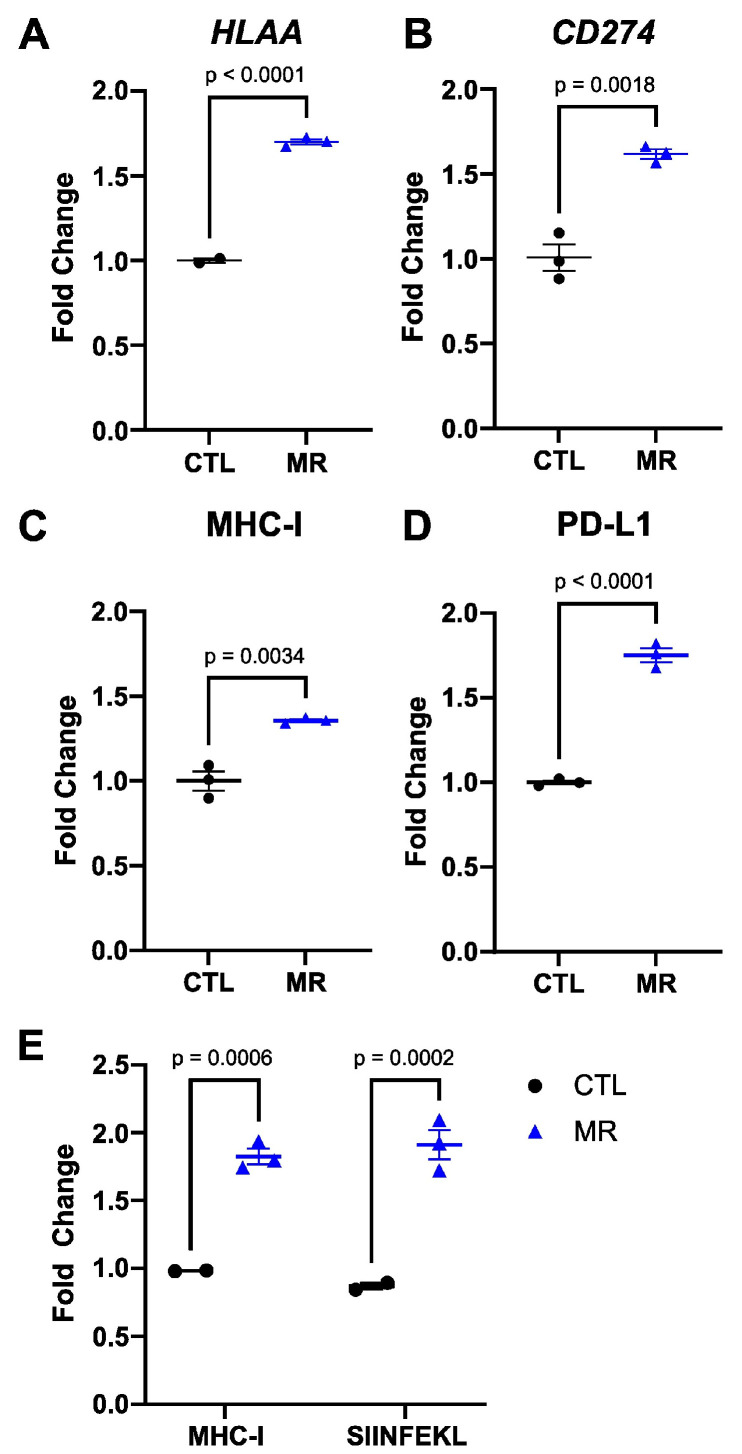
Methionine restriction increases MHC-I and PD-L1 expression. HT29 human colon cancer cells were grown for 24 h in media containing 200 µM (CTL) or 5 µM (MR) methionine. Gene expression for *HLAA* (**A**) and *CD274* (**B**) increased with MR. Surface abundance measured by flow cytometry (**C**,**D**) also increased with MR. In B16-OVA cells, lowering methionine also increased MHC-I surface abundance and the presentation of the SIINFEKL peptide (**E**). *t*-test with mean and SEM is shown.

**Figure 3 cancers-15-04467-f003:**
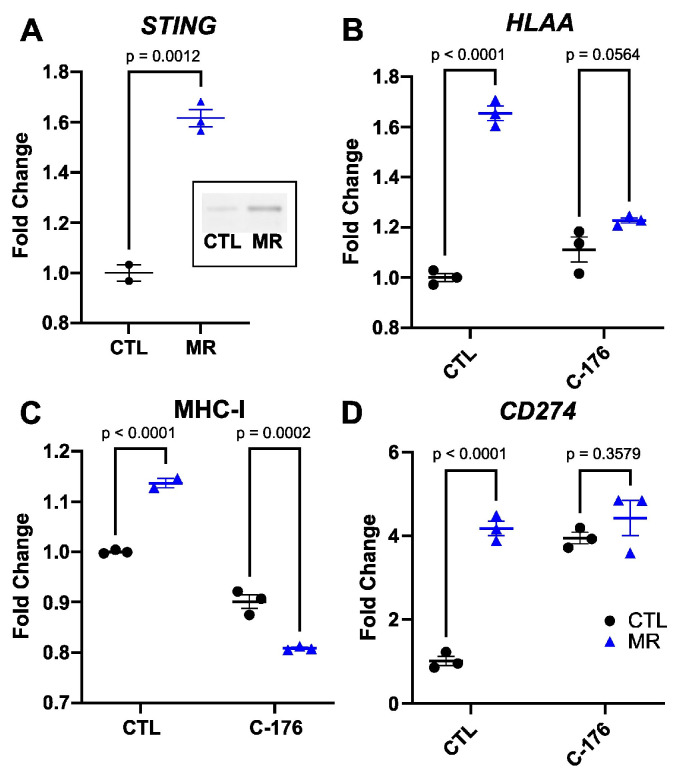
Inhibiting STING blunts the increase in MHC-I. *STING* was increased by MR at the gene and protein expression levels (**A**) and insert, *t*-test with mean and SEM). Inhibiting *STING* with C-176 led to a reduction in the increase in *HLAA* gene expression (**B**) and in its cell surface representation (**C**). CD274 gene expression was increased by C-176 and did not increase further with MR (**D**). Two-way ANOVA with mean and SEM.

**Figure 4 cancers-15-04467-f004:**
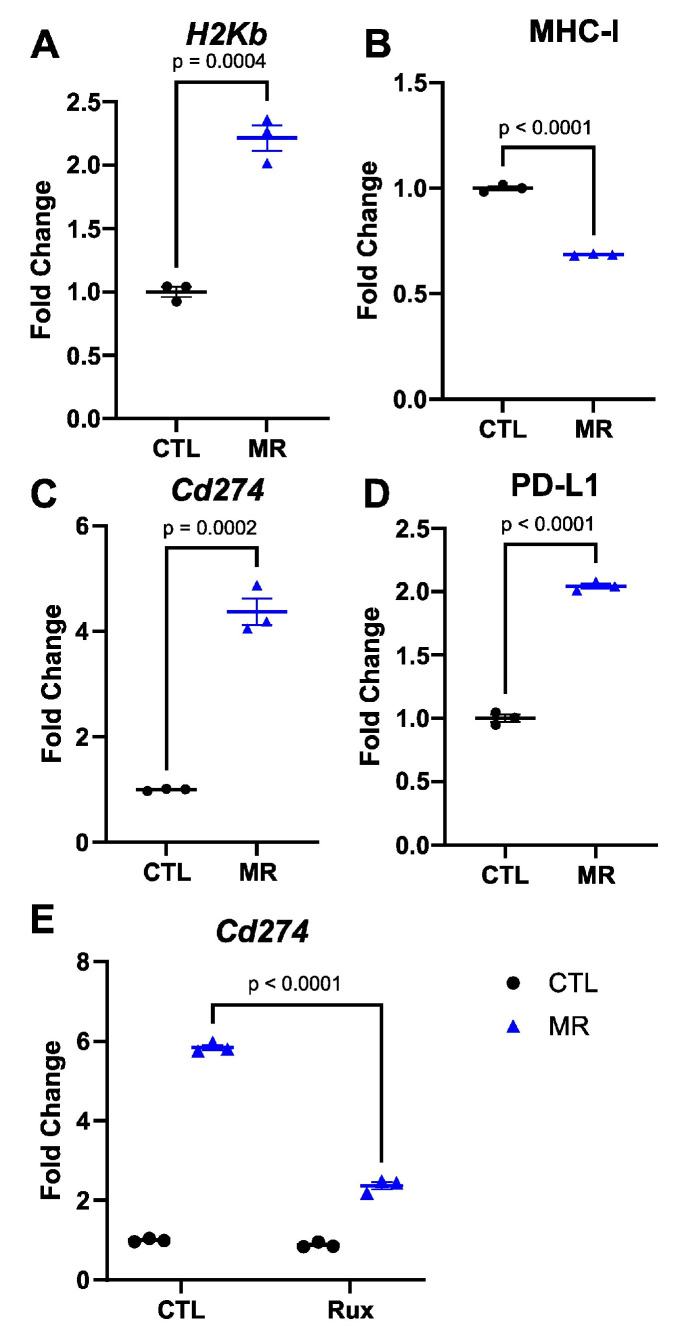
MHC-I and PD-L1 are altered by methionine restriction in a model cell line. MR increased *H2Kb* gene expression in vitro in the cell line MC38 (**A**). The cell surface expression of MHC-I, however, decreased with MR (**B**). The gene expression for *Cd274* also increased with MR (**C**), with a parallel increase in surface protein abundance (**D**). *t*-test with mean and SEM. Inhibiting JAK (interferon signaling) with 4 µM ruxolitinib led to a blunting of the effect of MR on Cd274 gene expression (**E**). Two-way ANOVA with mean and SEM.

**Figure 5 cancers-15-04467-f005:**
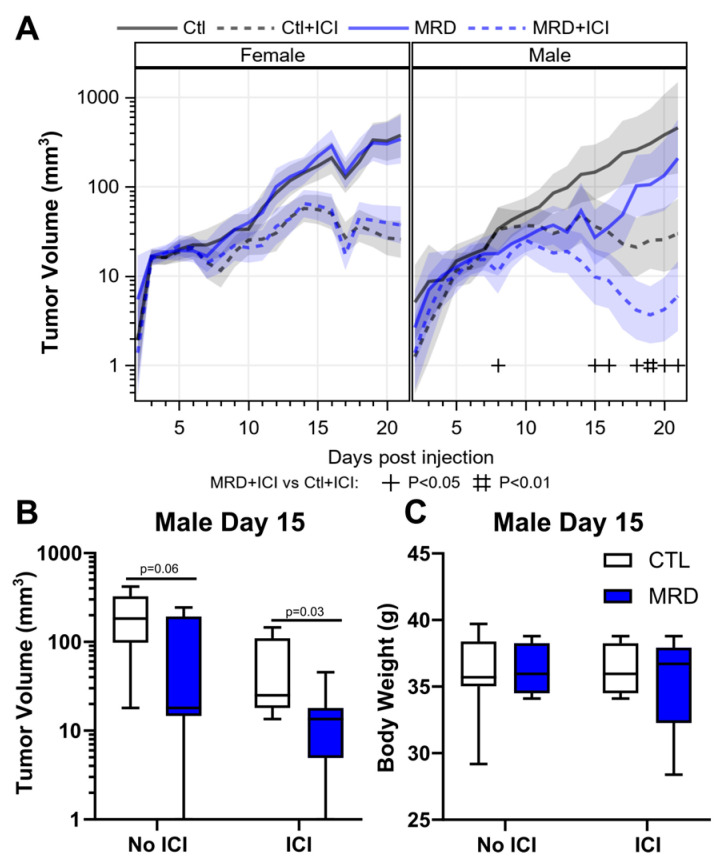
Dietary methionine restriction synergizes with ICIs in males. After subjecting the mice to a combination treatment consisting of anti-CTLA-4 and anti-PD-1 with MRD, females showed no benefit from MRD (**A**). In males, there was a 5× reduction in tumor volume with ICIs and MRD compared with ICIs alone (**B**) and (**C**). There was no significant change in body weight measured after 10 days of MRD (**C**).

**Table 1 cancers-15-04467-t001:** CPS for PD-L1 expression in MC38 tumors. CPS: PD-L1-stained cells/total tumor cells × 100.

Group	PD-L1 Score
CTL	15
CTL	7
CTL	4
CTL	3
CTL	2
MRD	13
MRD	10
MRD	18
MRD	15
MRD	20

## Data Availability

Data is contained within the article or Appendix A.

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
