# Peer review of "Increased Response to Immune Checkpoint Inhibitors with Dietary Methionine Restriction in a Colorectal Cancer Model"

_cancers, 2023, doi:10.3390/cancers15184467_

Round 1

Reviewer 1 Report

Dear authors,

I'm very happy to review your manuscript. I give you some comments to improve your manuscript as follows.

1.     Please highlight the novelty and the conclusion in short with a schematic of the “role of immune checkpoint inhibitors with dilatory methionine restriction in a CRC” this review in the introduction section.  

2.     Please add the proper format of the figures. If possible, please add your significant result in a graphical manner.

3.     Please more add. more about methodology.

4.     Please follow the appropriate format of journals with the appropriate reference and style.

Best regards

Author Response

The authors would like to thank the three reviewers for their comments and insights. Doing a bit of editorial work myself, I appreciate that it is often difficult to find reviewers and we are particularly grateful that three scientists took time from their busy schedule to do this important work for the scientific community. We hope that we addressed all comments to the reviewers’ satisfaction. The reviewers will find the changes indicated with blue text in the revised manuscript.

Reviewer 1:

Dear authors,

I'm very happy to review your manuscript. I give you some comments to improve your manuscript as follows.

  1. Please highlight the novelty and the conclusion in short with a schematic of the “role of immune checkpoint inhibitors with dilatory methionine restriction in a CRC” this review in the introduction section.

We thank the reviewers for the suggestion, and we added a schematic to summarize our findings at the end of the introduction.

  1. Please add the proper format of the figures. If possible, please add your significant result in a graphical manner.

Based on the comment from reviewer 2, I think the reviewer refers to the label at the top of the figures. I removed the labels in the figures. We also added the summary (comment #1).

  1. Please more add more about methodology.

We added information to our materials and methods about the western blotting protocol as well as the flow cytometry labels, compensations, and gating strategy.

  1. Please follow the appropriate format of journals with the appropriate reference and style.

We apologize for the formatting error and corrected the referencing style.

Reviewer 2 Report

The authors investigated the effect of methionine reduction on checkpoint inhibitor therapy in mice. This needs to be added to the title of the paper. The abstract summarizes the work well. Defined tumor cell lines and inbred mice were used. The relevance of the checkpoint molecules studied is described. The effect of methionine reduction on checkpoint molecules should be investigated. Material and methods are well described, but a lot of detailed information on cytometry is missing, here information according to MIFlowCyt needs to be added (doi: 10.1002/cyto.a.20623). The Data Availability statement is surprising and not correct. The results are well presented, Table 1 lacks the legend. The Discussion should clarify what the limitations of such a highly standardized model study in mice are.    

Author Response

The authors would like to thank the three reviewers for their comments and insights. Doing a bit of editorial work myself, I appreciate that it is often difficult to find reviewers and we are particularly grateful that three scientists took time from their busy schedule to do this important work for the scientific community. We hope that we addressed all comments to the reviewers’ satisfaction. The reviewers will find the changes indicated with blue text in the revised manuscript.

The authors investigated the effect of methionine reduction on checkpoint inhibitor therapy in mice. This needs to be added to the title of the paper. The abstract summarizes the work well. Defined tumor cell lines and inbred mice were used. The relevance of the checkpoint molecules studied is described. The effect of methionine reduction on checkpoint molecules should be investigated.

The molecular checkpoint PD-L1 was chosen here as it is expressed in tumor cells, which we could study in vitro to later translate in vivo. Other checkpoints are not significantly expressed on tumor cells. PD-1 is expressed on T-cells, CTLA-4 is expressed on T cells, and CD80/86 is expressed on APCs. This is why we chose to focus our analysis on PD-L1.

Material and methods are well described, but a lot of detailed information on cytometry is missing, here information according to MIFlowCyt needs to be added (doi: 10.1002/cyto.a.20623).

See comment 3 in reviewer 1. We added information to our materials and methods about the western blotting protocol as well as the flow cytometry labels, compensations, and gating strategy.

The Data Availability statement is surprising and not correct.

We apologize for the mistake and removed this section

The results are well presented, Table 1 lacks the legend.

We added a legend for Table 1 and reworked it to improve the clarity.

The Discussion should clarify what the limitations of such a highly standardized model study in mice are.   

We are now concluding on that remark and stressing the need for conformation in humans.

Reviewer 3 Report

      In this research, the authors revealed the increased response to immune checkpoint inhibitors with dietary methionine restriction in colorectal cancer. In my opinion, the current version of this manuscript fits the scope of Cancers and could be accepted after major revision.

My specific comments are in detail listed below:

1.     In Line 61-67, the current development or the newly revealed role of PD-L1 expression in speeding up the DNA damage repair process should be added in this part. Some references should be added to this part including 10.1002/advs.202207608.

2.     In all the figures, some spare parts like Figure 1, Figure 2, and et al. were shown. In my opinion, these parts should be removed.

3.     In this study (Line 360-368), how the metabolism of dietary methionine affects the status of PD-L1 should be discussed. Besides, what’s the mechanism behind it? Some references should be added to this part including 10.1002/adma.202206121.

4.     Some minor mistakes exist in the references, such as Ref. 18. The authors should correct it.

5.     Some references are out of date. Some new recent references may be better.

6.     In the conclusion part, the references should be removed.

Author Response

The authors would like to thank the three reviewers for their comments and insights. Doing a bit of editorial work myself, I appreciate that it is often difficult to find reviewers and we are particularly grateful that three scientists took time from their busy schedule to do this important work for the scientific community. We hope that we addressed all comments to the reviewers’ satisfaction. The reviewers will find the changes indicated with blue text in the revised manuscript.

In this research, the authors revealed the increased response to immune checkpoint inhibitors with dietary methionine restriction in colorectal cancer. In my opinion, the current version of this manuscript fits the scope of Cancers and could be accepted after major revision.

My specific comments are in detail listed below:

  1. In Line 61-67, the current development or the newly revealed role of PD-L1 expression in speeding up the DNA damage repair process should be added in this part. Some references should be added to this part including 10.1002/advs.202207608.

We are now commenting on the involvement of PD-L1 in DNA repair and cited the suggested reference.

  1. In all the figures, some spare parts like Figure 1, Figure 2, and et al. were shown. In my opinion, these parts should be removed.

We removed the superfluous figure labels.

  1. In this study (Line 360-368), how the metabolism of dietary methionine affects the status of PD-L1 should be discussed. Besides, what’s the mechanism behind it? Some references should be added to this part including 10.1002/adma.202206121.

We added more information about the role of type II interferon in regulating PD-L1 expression. We are also referring to the data presented in Supplementary figure 2 about Ifngr and Stat1 in supporting that statement.

  1. Some minor mistakes exist in the references, such as Ref. 18. The authors should correct it.

We re-formatted the references and added the page number for reference #18.

  1. Some references are out of date. Some new recent references may be better.

Some of the pioneering work we are basing our study upon was performed in the 1990s and 2000s, justifying the need to cite these studies. We have added some recent references suggested by the reviewers in comments 1 and 3. We have indicated the new references in the manuscript.

  1. In the conclusion part, the references should be removed.

We agree with the reviewer. We moved the referenced sentences to the discussion and the take-home points to the conclusion.

Round 2

Reviewer 2 Report

Thank you very much for considering reviewers' comments-

Reviewer 3 Report

The current version of this manuscript could be accepted.